# Wiedemann–Steiner Syndrome with a Pathogenic Variant in *KMT2A* from Taiwan

**DOI:** 10.3390/children8110952

**Published:** 2021-10-22

**Authors:** Chung-Lin Lee, Chih-Kuang Chuang, Huei-Ching Chiu, Ru-Yi Tu, Yun-Ting Lo, Ya-Hui Chang, Hsiang-Yu Lin, Shuan-Pei Lin

**Affiliations:** 1Department of Pediatrics, MacKay Memorial Hospital, Taipei 10449, Taiwan; clampcage@gmail.com (C.-L.L.); g880a01@mmh.org.tw (H.-C.C.); wish1001026@gmail.com (Y.-H.C.); 2Institute of Clinical Medicine, National Yang-Ming Chiao-Tung University, Taipei 11221, Taiwan; 3Department of Medicine, MacKay Medical College, New Taipei City 25245, Taiwan; 4MacKay Junior College of Medicine, Nursing and Management, Taipei 11260, Taiwan; 5Department of Rare Disease Center, MacKay Memorial Hospital, Taipei 10449, Taiwan; andy11tw.e347@mmh.org.tw; 6Division of Genetics and Metabolism, Department of Medical Research, MacKay Memorial Hospital, New Taipei City 25160, Taiwan; mmhcck@gmail.com (C.-K.C.); likemaruko@hotmail.com (R.-Y.T.); 7College of Medicine, Fu-Jen Catholic University, Taipei 24205, Taiwan; 8Department of Medical Research, China Medical University Hospital, China Medical University, Taichung 40402, Taiwan; 9Department of Infant and Child Care, National Taipei University of Nursing and Health Sciences, Taipei 11219, Taiwan

**Keywords:** *KMT2A*, Wiedemann–Steiner syndrome, whole-exome sequencing, Taiwan

## Abstract

Wiedemann–Steiner syndrome (WSS) is a rare genetic disorder. Patients with WSS have characteristics of growth retardation, facial dysmorphism, hypertrichosis cubiti (HC), and neurodevelopmental delays. WSS is in an autosomal dominant inherited pattern caused by a mutation of the *KMT2A* gene (NM_001197104.2). In this article, we discuss a 5-year-old boy who has mild intellectual disability (ID), hypotonia, HC, hypertrichosis on the back, dysmorphic facies, psychomotor retardation, and growth delay. Trio-based whole-exome sequencing (trio-WES) was carried out on this patient and his parents, confirming the variants with Sanger sequencing. Trio-WES showed a de novo mutation of the *KMT2A* gene (NM_001197104.2: c.4696G>A, p.Gly1566Arg). On the basis of the clinical features and the results of the WES, WSS was diagnosed. Therefore, medical professionals should consider a diagnosis of WSS if patients have growth retardation and development delay as well as hirsutism, particularly HC.

## 1. Introduction

Wiedemann–Steiner syndrome (WSS) (OMIM: #605130) is an autosomal dominant genetic disease. It was first described by Wiedemann et al. in 1989 [1], and Steiner and Marques defined WSS as a syndrome in 2000 [2]. Patients with WSS are usually described as possessing the following characteristics: hypertrichosis cubiti (HC); hypertrichosis on the back; short stature; psychomotor retardation; growth delay; small and puffy hands; and dysmorphic facies, including thick and arching eyebrows and down-slanting palpebral fissures. In intellectual disability (ID) patients, WSS is a main cause [3]. 

WSS is caused by the mutation of the *KMT2A* gene (lysine methyltransferase 2A, also known as the *MLL* gene) (NM_001197104.2), which was found in 2012 by Jones et al. [4]. The *KMT2A* gene (NM_001197104.2) is located on chr11q23.3 and encodes a histone methyltransferase (HMT) enzyme. This enzyme regulates the gene expression profile in early embryonic development and hematopoiesis [5]. To date, 266 public variants of *KMT2A* have been reported in the Leiden Open Variation Database (https://databases.lovd.nl/shared/genes/KMT2A) accessed on 5 July 2021 [3,6,7,8,9,10,11,12,13,14]. In the Human Gene Mutation Database (HGMD) (http://www.hgmd.cf.ac.uk/ac/gene.php?gene=KMT2A) accessed on 5 July 2021, 163 variants have been documented. More patients have been diagnosed with WSS using whole-exome sequencing (WES).

Until now, the complete WSS phenotype has not been fully understood. In this article, we discuss a 5-year-old Taiwanese boy with the clinical features of WSS using trio-based WES (trio-WES) to identify a de novo missense pathogenic variation of the *KMT2A* gene.

## 2. Case Report

This patient was born at 38 weeks’ gestation in a normal, spontaneous delivery, with a birthweight of 3225 g (29th percentile, *Z* score = −0.55), a body length of 48 cm (23rd percentile, *Z* score = −0.75), and a head circumference of 35 cm (34th percentile, *Z* score = −0.42). The patient’s parents were non-consanguineous, and they were phenotypically normal. The boy was referred to our genetic clinic due to mild ID and growth delay. At our clinic, the patient’s weight, and height were 16.6 kg (20th percentile, *Z* score = −0.83), 104.9 cm (19th percentile, *Z* score = −0.82), respectively. Physical examination showed dysmorphic facies and thick hair (Figure 1a,b); hypertrichosis on the back, arms, and lower limbs but not on the abdomen; and HC. The dysmorphic facies included a mildly coarse face with mild mid-face hypoplasia, mild trigonocephaly, an always-open mouth with a mild tongue protrusion, and hypertelorism. He had no feeding problems, vertebral block, sacral dimple, and renal or cardiac malformations. No other family members have similar symptoms.

We collected peripheral blood from the patient and his parents for genomic DNA at the age of 5 years. Trio-WES was carried out by the National Health Research Institutes (Taipei, Taiwan). They used the Illumina HiSeq Xten system. The trimmed reads were mapped to the reference human genome (hg19). We identified the variants using the following databases: the Single Nucleotide Polymorphism Database (https://www.ncbi.nlm.nih.gov/snp/) accessed on 5 July 2021, ClinVar (https://www.ncbi.nlm.nih.gov/clinvar/) accessed on 5 July 2021, the Genome Aggregation Database (http://gnomad.broadinstitute.org/) accessed on 5 July 2021, the Exome Aggregation Consortium (http://exac.broadinstitute.org/) accessed on 5 July 2021, the 1000 Genomes Project (http://browser.1000genomes.org) accessed on 5 July 2021, and HGMD (http://www.hgmd.cf.ac.uk/) accessed on 5 July 2021. We confirmed the identified variants via Sanger sequencing and interpreted the variants using the guidelines of the American College of Medical Genetics and Genomics (ACMG) [15].

We identified a missense variant with heterozygous state in the *KMT2A* gene (NM_001197104.2: c.4696G>A, p.Gly1566Arg), which we confirmed via Sanger sequencing (Figure 2a,b). The patient’s parents did not have this variant and it indicated a de novo variant. This variant was reported in ClinVar and was classified as a likely pathogenic (ACMG criteria code: PS2+PP2+PP4). 

## 3. Discussion 

Wiedemann et al. and Steiner and Marques reported and defined WSS in 1989 and 2000, respectively [1,2]. Koenig et al. reported three additional cases in 2010 [16]. Different clinical phenotypes exist according to ethnicity. For example, in a French cohort, 65%, 61%, and 33% of 33 cases had eating difficulties, HC, and microcephaly, respectively [3]. In addition, eating difficulties, HC, and microcephaly were noted in 31%, 44%, and 50% of 16 cases, respectively, in a Chinese cohort [12]. HC is seen in most patients. However, there are reports of patients who do not have HC in the literature [3,4,13,17,18]. Neurodevelopmental disorders, including WSS, are related to variations of the *KMT2A* gene [7]. Molecular diagnoses such as WES, whole-genome sequencing, and targeted sequencing for the *KMT2A* gene could be useful for genetic counseling and patient care. Additional WSS cases could be found using WES, especially trio-WES [3,12,19,20].

The major variants of the previously described WSS cases were de novo null variants [3,8,12,13,20]. *KMT2A* gene missense variants made up a small but significant portion [3,7,12]. The ClinVar database (2021/02/09) reported 113 variants of the *KMT2A* gene associated with WSS. Among these variants, 67 are classified as pathogenic, 22 as likely pathogenic, 23 as of uncertain significance, 1 as likely benign, and none as benign. Furthermore, 32 are identified as frameshift small indels, 44 as missense mutations, 26 as nonsense mutations, 9 as splicing mutations, and 2 as antisense RNA. Because the exons 3 and exon 27 are much larger than the other exons, about 60% of the mutations are located in these regions. 

According to a previous study with a cohort of 33 French WSS cases [3], frameshift (41%), nonsense (28%), and missense (28%) were the most frequent types of mutation. In another study with a Chinese cohort of 16 patients [12], the results of the mutation spectrum were like those of the French cohort. In our study, we present a 5-year-old Taiwanese boy with a de novo missense variant of *KMT2A* (NM_001197104.2: c.4696G>A, p.Gly1566Arg). His symptoms, which are compatible with WSS, include mild ID; growth delay; dysmorphic facies and thick hair; and hypertrichosis on the back, arms, and lower limbs but not on the abdomen.

HMT is encoded by the *KMT2A* gene. It is an important gene expression regulator in early development. HMT regulates multiple Wnt-related and Hox-related genes in histone H3 lysine 4 methylation [21]. Due to the function of HMT, the phenotypes of patients with WSS are complex. The phenotypes involve multiple systems, such as organic problems, developmental delay, ID [3,16], distinct craniofacial features, and skeletal anomalies [22].

Patients with language delay and ID require special education classes with regular speech and occupational therapies. If symptoms of hypotonia are noted, patients should have carnitine supplementation (in our patient, 2018/11/19: Plasma Free Carnitine: 40.9 μmol/L, normal range: 30~45.4 μmol/L; Plasma Total Carnitine: 49.9 μmol/L, normal range: 36.6~54 μmol/L). For patients with cardiac anomalies, surgical treatment should be considered. Because of growth retardation, children with WSS should have their pituitary function investigated. If biochemical evidence of growth hormone deficiency is noted, a pituitary magnetic resonance imaging scan should be considered, and patients should receive recombinant human growth hormone treatment. All these treatments can help patients with WSS survive with a better quality of life. Table 1 shows the recommended multi-disciplinary plan of care for individuals with WSS.

WES, a powerful tool used to identify disease-causing variants, allows the relationship between genotype and phenotype to become clearer. Shashi et al. [23] stated that next-generation sequencing should be applied if a diagnosis has not been confirmed after using more traditional approaches. Patients with genetic disorders, who often experience considerable psychosocial and economic burdens, can benefit from WES due to the increased rate of genetic disorders and cost savings. WSS prevalence would be updated if more patients were diagnosed using WES.

## 4. Conclusions

In summary, we discussed a 5-year-old Taiwanese boy who was diagnosed with WSS via clinical features and trio-WES. He is the first patient with WSS presented in Taiwan. WSS is a rare genetic disorder that combines with sporadic syndrome and phenotypic heterogeneity. Therefore, our findings can be valuable in genetic diagnosis and mutation-based screening. 

## Figures and Tables

**Figure 1 children-08-00952-f001:**
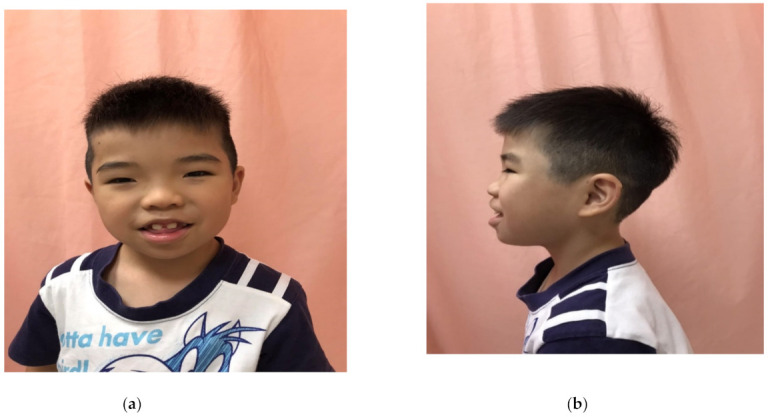
Physical appearance of a 5-year-old boy with WSS. Dysmorphic facies (**a**), depressed nasal bridge (**b**) and thick hair (**a**,**b**). Dysmorphic facies, such as mildly coarse face with mild mid-face hypoplasia, mild trigonocephaly, an always-open mouth with a mild tongue protrusion, and hypertelorism (**a**).

**Figure 2 children-08-00952-f002:**
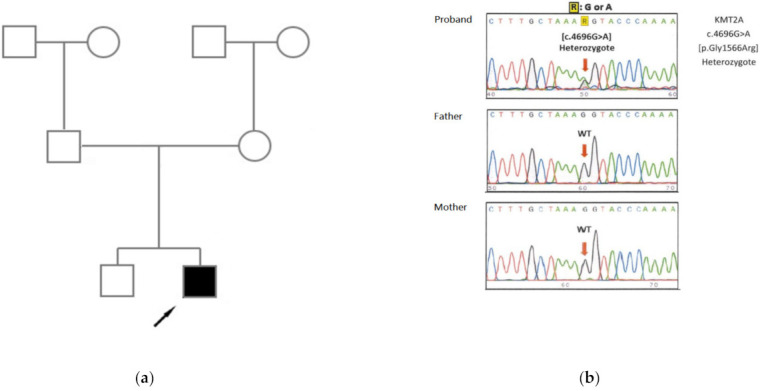
NM_001197104.2: c.4696G>A (p.Gly1566Arg) variation by Sanger sequencing. (**a**) Patient family tree. The blank squares and circle indicate unaffected males and females, respectively; the filled square indicates an affected male, and the arrow indicates the proband. (**b**) Electrophogram of heterozygous *KMT2A* c.4696G>A (red arrow). Both parents carried the wild-type sequence at this nucleotide.

**Table 1 children-08-00952-t001:** Recommended multi-disciplinary plan of care for Wiedemann-Steiner Syndrome.

Subspecialist	Considerations/Screening	Evaluation
Development	Language delay and intellectual disability	Regular speech and occupational therapies
Neurology	Hypotonia	Carnitine supplementation
Endocrinology	Growth retardation	Pituitary magnetic resonance imaging scan and recombinant human growth hormone treatment
Cardiology	Cardiac anomalies	Surgical treatment
Geneticist	*KMT2A* gene analysis and genetic counseling	Diagnostic discussion and family planning

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
