# Peer review of "Wiedemann–Steiner Syndrome with a Pathogenic Variant in KMT2A from Taiwan"

_children, 2021, doi:10.3390/children8110952_

Round 1

Reviewer 1 Report

Recommendations: 

  1. Title should be reconsidered: Avoid use of "First case" "Case report".
  2. I don't think you should said you did a "literature review" on your title as I don't see an extensive discussion about what is WSS. 
  3. This case report just present a new case of WSS from the novo. 
  4. Authors mention the use of carnitine in patient with WSS and hypotonia, but did't included carnitine profile on patient case description (if was available).
  5. Formatting of the actual table is difficult to follow. Please rearrange is a more organized way. 
  6. Picture included have extra space should be cropped out. Picture (B) should be replaced with a lateral view of the actual patient facial features like Depressed nasal bridge. Should include a picture of the Dorsal hypertrichosis  in a panel with the rest of the features. 
  7. The pedigree should include three generations in a publishable format. See: Bennett RL, French KS, Resta RG, Doyle DL. Standardized human pedigree nomenclature: update and assessment of the recommendations of the National Society of Genetic Counselors. J Genet Couns. 2008 Oct;17(5):424-33. doi: 10.1007/s10897-008-9169-9. Epub 2008 Sep 16. PMID: 18792771.
  8. Very poor resolution on the Genetic Electrophogram. Should be improved. 
  9. Different types of fonts are presented thru the article: Be consistent. 
  10. On the discussion authors presented an management approach for patients with WSS. I recommend to create a table to present the approach proposed or suggested. See the following paper to use as an example: Ramirez-Arenalde MA, Bruckman-Blanco WJ, Frontanes-Heredia A, Santiago-Castro SL, De Jesús-Rojas W. An Extremely Rare Case of Birk-Barel Syndrome With Severe Central Apneas. Cureus. 2021;13(6):e15862. Published 2021 Jun 23. doi:10.7759/cureus.15862

Reviewer 2 Report

This manuscript describes a patient with a de novo KMT2A gene variation in a patient with WSS. The variant is a known variant and syndrome is a well known situtation. Although the manuscript is well-writen , the variant found in the patient is a known variant and patient represents a common phenotype of the disease.

-Hyeprthrichosis cubiti is a part of clinical fetures of  Wiedemann Steiner syndrome but there are patients who do not have HC in the literature ([Jones et al., 2012; Aggarwal et al., 2017; Baer et al., 2018;
Ramirez-Montaño and Pachajoa, 2019 and Demir et al 2021).

-Percentages given in the table are summarized from which publications? Considering HPO (Human Phenotype Ontology) terms may be more efficent for clinical descriptions according to my personal opinion. 

Round 2

Reviewer 2 Report

The manuscript seems improved in its current form.